# Predicting speed of progression of lens opacification after pars plana vitrectomy with silicone oil

**Philipp Schindler** [1]☯*, **Luca Mautone**[1]☯, **Vasyl Druchkiv**[2], **Toam Katz**[1], **Martin Stephan Spitzer**[1], **Christos Skevas**[1]

1 Department of Ophthalmology, University Medical Center Hamburg-Eppendorf, Hamburg, Germany,
2 Department of Research & Development, Clínica Baviera, Valencia, Spain

☯ These authors contributed equally to this work.
* p.schindler@uke.de

## Abstract

### Purpose

An increasing number of posterior segment disorders is routinely managed with pars plana vitrectomy (PPV). In older, phakic patients cataract formation is expected within the first two years after surgery. For younger patients its progression is individually fluctuating. This study uses an objective quantitative measurement for lens-status-monitoring after PPV with silicone oil to derive predictions for progression and severity of post-operative lens opacification evaluated in patients with rhegmatogenous retinal detachment (RRD).

### Methods

Data acquisition was performed prospectively between March 2018 and March 2021. Penta-camHR® Nucleus Staging mode (PNS) was used to objectively gather data about nuclear cataracts after PPV at different time points. Data was grouped into training and test sets for a mathematical prediction model. Via backward variable selection method a mathematical formula was set up by means of which predictions about lens densitometry (LD) can be calculated.

### Results

20 males [58.8%] and 14 females [41.2%] matched the inclusion criteria (mean age 50.6 years [23–75; ±12.3]). Average follow-up was 8.1 months (3,4–17.4; ±3.4). Mean baseline LD of the treated and fellow eye before surgery was 11.1% (7.7%-17.6%; ±2.0) and 11.2% (7.7%-14.8%; ±1.5), respectively. Predicted LD values by the model for five pre-selected patients closely match the observed data with an average deviation of 1.06%.

### Conclusions

Using an objective parameter like LD delivered by the PentacamHR® PNS mode additionally to the patient's age allows us to make an individual prediction for any time after PPV with silicone oil due to RRD for all ages. The accuracy of the model was stronger influenced

**Data Availability Statement:** All relevant data is located at https://statisticarium.com/apps/sample-apps/LensDensityOil/.

**Funding:** The author(s) received no specific funding for this work.

**Competing interests:** The authors have declared that no competing interests exist.

by baseline LD as cofactor in the equation than patient's age. The application for the prediction lens opacification [which can be accessed for free under the following link (https://statisticarium.com/apps/sample-apps/LensDensityOil/)] can help vitreoretinal surgeons for patient consultation on the possibility to combine PPV with cataract surgery.

## Introduction

An increasing number of posterior segment disorders is successfully managed with pars plana vitrectomy (PPV). As a result vitrectomy related cataractogenesis is a common surgical side effect, which is expected within the first two years after surgery and it is more severe in elderly patients [1, 2]. For patients over 50 years of age it has already been shown several times that significant lens opacification mostly occurs within the first 2 years after PPV [3–5].

However, this is not the case for all patients receiving PPV. The exact biological mechanism that leads to the progression of the lens opacity is still vividly discussed and has not yet been definitely clarified. Postoperative elevated levels of oxygen and free radicals seem to play an important role [6–9].

It has already been shown for C3F8-gas that lens opacity after a PPV can be predicted for every point in time after surgery with sufficient accuracy by using lens densitometry with a mathematical model [10]. PPV nowadays is more and more often combined with cataract surgery in patients over 50 years of age, but it may also be considered in younger patients. However, removing the lens during the vitrectomy procedure also has disadvantages such as an additional risk of surgical complications related to phacoemulsification, increased postoperative inflammation, less predictable postoperative outcome than in sequential surgery and an increased risk of silicone oil spill-over in the anterior chamber. Thus, it may be desirable for advising patients and surgical planning to be able to predict on an individual patient basis how soon (if at all) cataract surgery after vitrectomy with silicone oil will become necessary.

In this study we evaluated patients of various age and lens densitometry values that were treated for rhegmatogenous retinal detachment (RRD) via the aforementioned regimen. To receive a quantitative analysis of lens transparency the PentacamHR® Nucleus Staging (PNS) module of the Scheimpflug tomography system (Oculus, Wetzlar, Germany) was used.

## Methods

In our prospective Study patients with RRD treated with 23G PPV using silicone oil as an endotamponade from March 2018 to March 2020 were included. The study followed the tenets of the Declaration of Helsinki and was approved by the Medical Institutional Review Board of Hamburg (PV7250). Patient's data were gathered fully anonymized. Written consent was waived for this study, but this was obtained verbally.

All patients underwent a complete ophthalmological examination with BCVA, intraocular pressure measurement, slit lamp biomicroscopy, dilated funduscopy, and most importantly PentacamHR® lens densitometry prior to surgery and at every visit during follow-up as it is standard procedure in our clinic. Post-operative follow-up visits were scheduled at about six weeks, three months and six months. If additional visits were necessary due to disease related further clinical controls, these measurements were also used for the analysis. Furthermore, there should not have been any complications during PPV that could accelerate lens opacification.

In addition, only patients were included for whom the surgeon had previously decided not to perform a combined phakovitrectomy.

PentacamHR® measurements were taken after pupil dilation and mean lens densitometry (LD) value was calculated by the PNS module in predefined three-dimensional volumes centered on the apex. The measurement and calculation is performed automatically within less than 3 seconds, provided the PNS module is installed on the device. All LD measurements have been performed by trained medical-laboratory assistants under standard dim-light conditions. The advantage of this system is that the parameters are collected from a three-dimensional portion of the lens nucleus and thus LD is calculated from a volume and not just from a linear section. This way it reflects a better overall approximation of the actual lens opacification. The mean lens densitometry is output as a percentage of total backward scatter of light.

Only PentacamHR® images without reflections or distortions were used for the evaluation. This method has been successfully established by prior various studies [10–16].

Patients with prior cataract surgery on one eye, PPV, or any other intraocular procedure were excluded. Patients with history of ocular trauma, uveitis, topical or systemic corticosteroid therapy, and signs of visually impairing cataract at baseline evaluation were also excluded from the analysis. Additionally, known topical or systemic conditions (diabetes excluded) that could accelerate cataract formation and/or progression after PPV led to exclusion from the study. Patients with ischemic and/or proliferative retinopathy were also excluded [17–19].

The need for cataract surgery during follow-up was marked as the endpoint for this case. LD measurements of fellow eyes were taken to provide reference values for LD progress and to show their natural course. The surgery was performed by 3 vitreoretinal surgeons with at least 3 years of post-fellowship experience in the field of retinal detachment. For details about the surgical technique we refer to the supplemental content.

We selected the following variables to co-evaluate with post-surgical LD changes:

- time after surgery (months)

- baseline LD (%)

- age at the time of surgery (years)

The sample was randomly divided into training and test data sets. Backward variable selection method based on Akaike information criterion was applied to develop a mixed regression model for prediction of LD. The observed LDs from the test data set were compared with the predicted LDs.

## Results

Thirty-four eyes of 34 patients matched the inclusion criteria. Demographic and presurgical parameter are summarized in Table 1.

Datasets of 20 males [58.8%] and 14 females [41.2%] were investigated. Seventeen patients each underwent surgery in the right or left eye. Patient's age ranged from 23 years to 75 years with an average of 50.6 (±12.3) years. Mean baseline LD of the treated and fellow eye were 11.1% (7.7%-17.6%; ±2.0) and 11.2% (7.7%-14.8%; ±1.5), respectively.

Average follow-up was 8.1 months (3.4–17.4; ±3.4). Three patients suffered from diabetes type II, but none of these patients had signs of diabetic retinopathy during the period of investigation.

### Statistics and prediction model

A backward variable selection method was applied starting with a saturated model with all interaction effects between time, age, baseline lens densitometry. The resulting fixed effect part of the model is:

**Table 1. Demographics and preoperative parameter.**

| Patients | Total n = 34 |
|---|---|
| Right eye | n = 17 (50.0%) |
| Left eye | n = 17 (50.0%) |
| Female | n = 14 (41.2%) |
| Male | n = 20 (58.8%) |
| Age (years) | 23–75; 50.6 (±12.3) |
| BCVA LogMAR [Snellen] | 0.0 [20/20]– 2.4 [hand motion]; 1.3 (±0.9) |
| Lens Densitometry baseline treated eye (%) | 7.7–17.6; 11.1 (±2.0) |
| Lens Densitometry baseline fellow eye (%) | 7.7–14.8; 11.2 (±1.5) |
| Diabetes Type I | n = 0 (0.0%) |
| Diabetes Type II | n = 3 (8.8%) |
| Follow-up time (months) | 3.4–17.4; 8.1 (±3.4) |

Qualitative parameters are presented as percentages, and quantitative ones as min to max and mean (±Standard deviation).

$$\text{LogitDensitometry} = \beta_0$$

$$\beta_1 \times \text{Months} +$$

$$\beta_2 \times \text{Age} +$$

$$\beta_3 \times \text{Baseline LD} +$$

$$\beta_4 \times \text{Months} \times \text{Age} +$$

$$\beta_5 \times \text{Months} \times \text{Baseline LD} +$$

$$\beta_6 \times \text{Age} \times \text{Baseline LD} +$$

$$\beta_7 \times \text{Months} \times \text{Age} \times \text{Baseline LD}$$

The supplement document of this paper gives further information on statistical calculations, prediction of LD, surgical method and results of the fellow eyes.

The model delivers a set of coefficients that describe the interaction between dependent factors, for example "months" after surgery or "baseline LD", and the prediction. The factors have been selected according to their significant influence on lens opacification in the statistical analysis.

There is a significant effect of time (months after surgery). But the effect of time enters the model in four ways: as a linear effect, an interaction with age, an interaction with baseline LD and higher order interaction with age and baseline LD. That explains why "months" appears several times and in different ways in our equation further below. Note that quadratic term of time was not significant, because there is no strong tendency to decreasing rate of change in the analysed period.

Furthermore, baseline LD is positively related to postoperative LD, meaning that all being equal the higher baseline LD is, the higher the predicted trajectories will be.

The effect of age is also significant, but not as much as baseline LD. Therefore, baseline LD is a higher influential cofactor than age. In the model we consider age as continuous variable.

All these findings were considered when calculating the coefficients $\beta_0$ through $\beta_7$. In this way we received the results summarized in Table 2.

Note that:

**Table 2. Mixed effects regression model parameters β₀ through β₇.**

| | Dependent variable: $\frac{\frac{LD}{100}}{1-\frac{LD}{100}}$ |
|---|---|
| **β₁ Months** | -0.567*** (0.143) |
| **β₂ Age** | 0.018 (0.025) |
| **β₃ Baseline lens densitometry** | 0.164 (0.133) |
| **β₄ Months x Age** | 0.013*** (0.003) |
| **β₅ Months x Baseline lens densitometry** | 0.055*** (0.016) |
| **β₆ Age x Baseline lens densitometry** | -0.002 (0.003) |
| **β₇ Months x Age x Baseline lens densitometry** | -0.001*** (0.0003) |
| **β₀ Constant** | -3.760*** (1.277) |
| **Observations** | **104** |
| **Log Likelihood** | 52.242 |
| **Akaike Inf. Crit.** | **-84.483** |
| **Bayesian Inf. Crit.** | -58.039 |

P-Values are given symbolically with asterisks:

*p<0.1;

**p<0.05;

***p<0.01. In parenthesis are standard errors of the coefficients.

- β₄ is positive → the older the patient the steeper (stronger) is the change.

- β₅ is positive → the higher the baseline LD the steeper is the change.

We see that time does not significantly enter the model. Both age and baseline LD seem to be significantly related to the LD. Since time is not related to the trajectories the prediction would be a horizontal line that only goes up or down depending on baseline LD and age. The model works by inserting values for each dependent variable into the following equation:

$$
\begin{aligned}
\text{Logit(LD)} = \quad & -3.760 \\
& -0.567 \times \text{Months} + \\
& 0.018 \times \text{Age} + \\
& 0.164 \times \text{Baseline LD} + \\
& 0.013 \times \text{Months} \times \text{Age} + \\
& 0.055 \times \text{Months} \times \text{Baseline LD} + \\
& -0.002 \times \text{Age} \times \text{Baseline LD} + \\
& -0.001 \times \text{Months} \times \text{Age} \times \text{Baseline LD}
\end{aligned}
$$

Where,

$$
Logit(LD) = \frac{\frac{LD}{100}}{1-\frac{LD}{100}}
$$

To get the final prediction we have to back-transform from the logit scale as follows:

$$LD = 100 \ \times \ \frac{e^{Logit(LD)}}{1 + e^{Logit(LD)}}$$

By filling the equation with individual values for "months" after surgery, "baseline LD" and "Age" of the Patient one can calculate and therefor predict an estimated LD for any time after surgery for any patient facing a PPV with silicone due to RRD.

Fig 1 shows the observed post-surgical LD trajectories of just 29 cases, since 5 cases have been excluded from the model creation process to test its power afterwards.

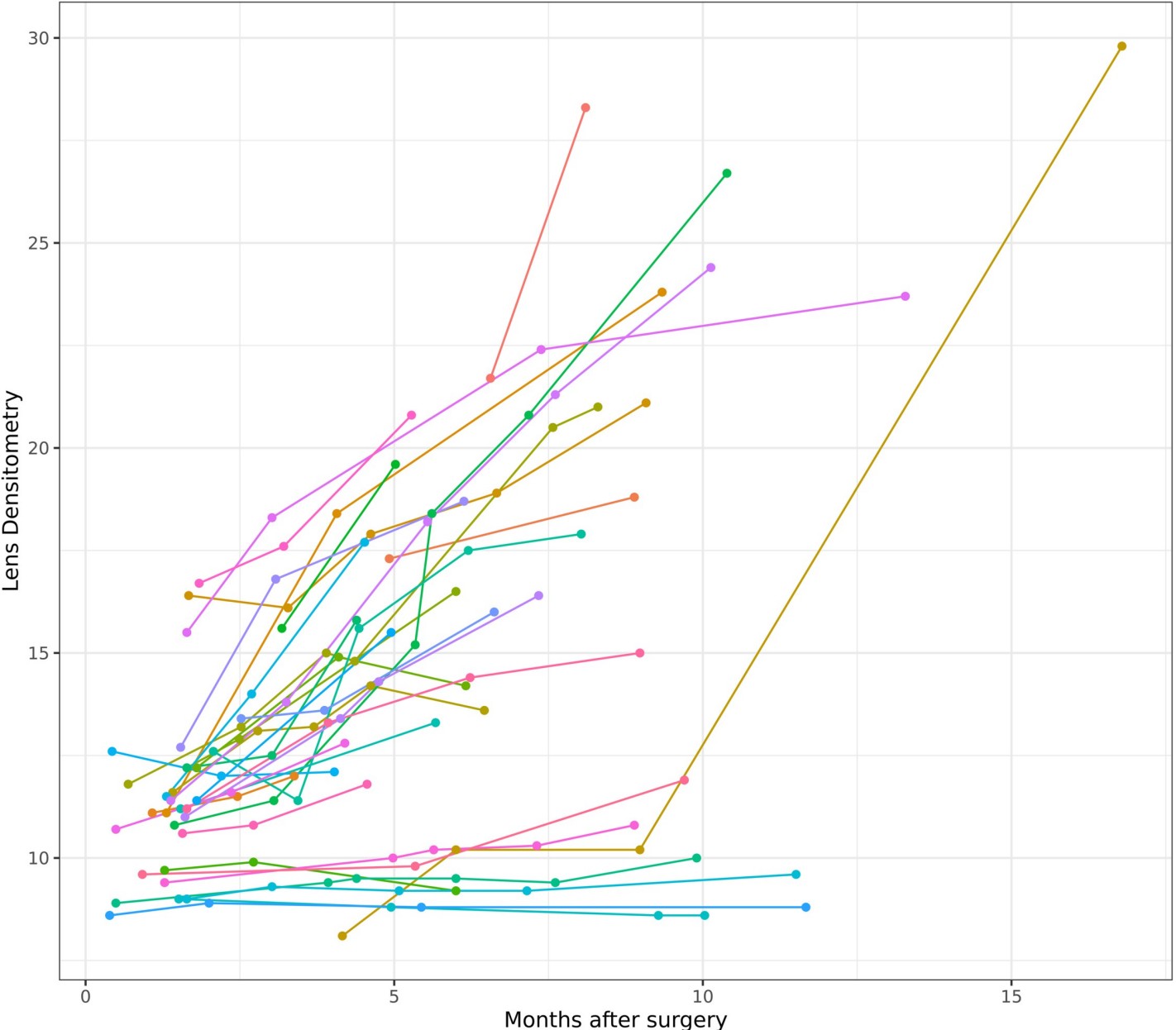

**Fig 1. Changes in trajectories of lens densitometry.** Trajectories of lens densitometry (%) during months after surgery for treated eyes.

Our statistical analysis has shown that age of the patient and baseline LD measurement are positively related, meaning that the older a patient is, the higher is his or her baseline LD. When looking at Fig 1 there appears to be two groups of patients. One group does not increase in LD over a period of 10 months. This group has in common that its baseline LD is below 10%. The range of age in this group is 23–44 years. Above 10% baseline LD all cases experience a rapid increase in LD within the investigated period. One of the younger patients with initially low baseline LD developed a steep increase in LD only after 10 months, although there was no notable increase in LD up to 10 months after surgery.

### Model performance

Now we used these five patients that have been excluded in advance for validation purpose.

In Fig 2 blue dots show predicted LD values from the model. Red dots depict actual LD values from the patient's PentacamHR® file, respectively.

Apart from ID = 5, blue and red dots are very close together. The prediction ends when there were no more recordings in the patient's file, but predictions can be done for any time after surgery for each individual. To prove that the prediction is working even for a period after 10 months one must look at ID = 5 again. His or her last LD measurement was made after 12 months and matches the predicted LD perfectly. This patient is 27 years of age and had a baseline LD of 8,3%.

**Web application for predicting postoperative lens densitometries.** As we did for PPV with C3F8-gas previously [10], we provide a web application that calculates individual postoperative trajectories for LD based on the model shown above. Only "age" in years and "baseline LD" must be provided by the user and the application displays the results as a line plot. The application can be accessed free under the following link: https://statisticarium.com/apps/sample-apps/LensDensityOil/.

## Discussion

Cataract formation is the most frequent complication after PPV in phakic eyes. But precise estimations when and to what degree it will occur are only vaguely possible so far. The cause for lens opacification may relate to increased oxygen levels and the ensuing oxidative stress, as well as the altered biochemistry in the vitreous cavity following vitreous removal [6, 7, 20].

Previous studies underlined that oxidative stress to the lens, for example by hyperbaric oxygen therapy or removal of the vitreous body, increases the exposure to molecular oxygen and can cause lens opacification [21–24].

Among other factors, age plays a key-role in causing not only age-related cataract, but also postsurgical lens opacification. For patients >50 years of age combined PPV is performed more frequently, but it is sometimes needed in younger patients, too. Kataria et al. and Thompson et al. suggest that NSC progresses at a rate 6- to 9-fold higher in patients >50 years of age compared with patients <50 years of age. This finding suggests that cataract surgery should not be performed routinely combined with PPV in patients less than 50 years of age because cataract surgery is usually not needed for many years [19, 23, 25–28].

Whether combined cataract and PPV surgery should be generally performed for patients over 50 years of age has not yet been clearly recommended.

The rate of lens opacification is simply too difficult to estimate for individual cases without objective parameters. Especially for the age group between 41–55 years there is no reliable evidence about how long it takes until a significant lens opacification occurs after PPV. Individual fluctuations are too wide in this regard. Using silicone oil as an endotamponade may lead to significantly earlier cataract formation after PPV than air or

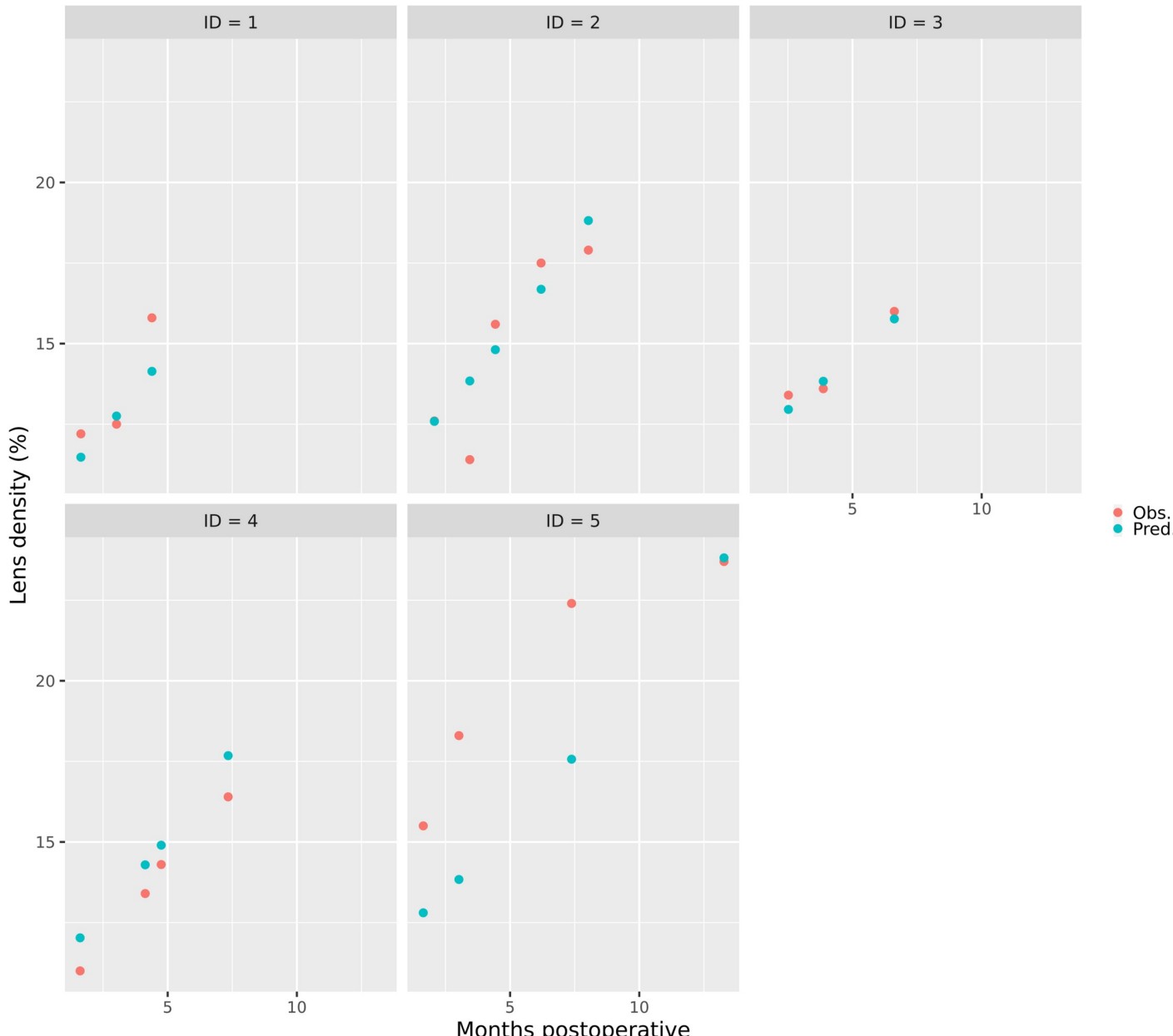

**Fig 2. Predicted and observed lens densitometries over time.** Predicted (blue dots) and observed (red dots) postoperative lens densitometries (%) over time (months after surgery).

gas. This finding is supported when looking at the data of the average LD 6 months after surgery from our previous study on the progression of lens opacity after PPV with C3F8 gas. With silicone oil mean LD of the cohort increases by approximately 5.5%, whereas under C3F8 gas {(n = 34; age (years) 32–77; 58.5 (±4.3); baseline LD treated eye (%) 8.7–14.8; 10.9 (±0.8)} there was only an increase of about 2.7% within 6 months after PPV [10].

Moreover, posterior subcapsular cataract (PSC) is more likely to develop in silicone oil filled eyes [29].

But based on the endotamponade used alone, it cannot be concluded when exactly vision impairing cataract will occur. Other factors, such as the surgeon's experience, sometimes play a role as well. For instance, Xu et al. were able to show that the time between PPV and cataract surgery was significantly longer for surgeons with more experience than for inexperienced surgeons [30].

As has already been shown in other studies, lens opacification progressed relatively fast after surgery in most of our cases. But LD trajectories in younger patients were flat and lens opacification progressed slower compared to older patients (see Figs 1 and 3) for at least 10–12 months following PPV. The fact that baseline LD of older people is naturally higher was considered when developing our model. However, the increase in LD with age did not always follow a linear fashion. Patients with higher age may have a relatively low baseline LD and younger patients may have an increased LD. Thus, baseline LD might be a better indicator than age for the decision whether a combined phakovitrectomy should be performed or not.

This assumption becomes clear when looking at the trajectories of patients who are under 51 years of age (see Fig 3). In the case of the 44-year-old patient with a baseline LD of 10.1%, LD does not increase significantly over time. In contrast, the 39-year-old patient with a baseline LD of 11,0% shows a significant increase in LD after just a few months. These two examples show that age alone is not a reliable parameter for estimating possible lens opacification after PPV. So, taking baseline LD into account and giving it a higher weight in the model than age, makes the difference when predicting the speed of progression of lens opacification after PPV.

When looking at Fig 3 there appear to be 2 groups in this study. One group which shows little or no increase in LD over a longer period of time and a second group which shows a significant increase shortly after PPV. Most of the patients in the group with a rapid increase in LD are over 51 years of age, but not all of them. On the other hand, one can see that the group with a slow increase in LD also includes patients who are at least over 40 years old. Individual fluctuations that cannot be attributed solely to the patient's age also seem to be present in this study. It remains unclear why some lenses opacify faster than others. However, younger lenses seem to be more stress-resistant to external influences such as intraocular operations or increased oxygen levels.

Various lens opacity grading systems exist for grading cataracts. The Lens Opacity Classification Systems II and III (LOCS II and III) use color lens photograph standards or slit-lamp photographs of the lens to quantify its opacity [31]. However, the LOCS grading systems are rather subjective and depend on human raters.

Using a rotating Scheimpflug camera PentacamHR® provides precise, three-dimensional images of the lens. It delivers reproducible data related to quantify LD and is an objective method to assess the lens transparency [12–14, 32–35].

Limitations are that the PentacamHR® PNS module only depicts nuclear and no cortical or PSC changes. PSC occurrence is described frequently under silicone oil and leads to considerable visual impairment quickly [36].

However, since PSC is transient in most cases and is usually followed by NSC after PPV with silicone oil we do not consider this weakness as a negative influence on the results of our study [37].

The cohort is also very homogeneous, as patients with previous intraocular surgeries or lesions were excluded, which limits the transferability of the results to a larger population.

Using age and baseline LD as individual factors to predict LD for any time after surgery showed precise results (Fig 2). Presumably, the accuracy of the prediction model will improve when more data is used to train the model.

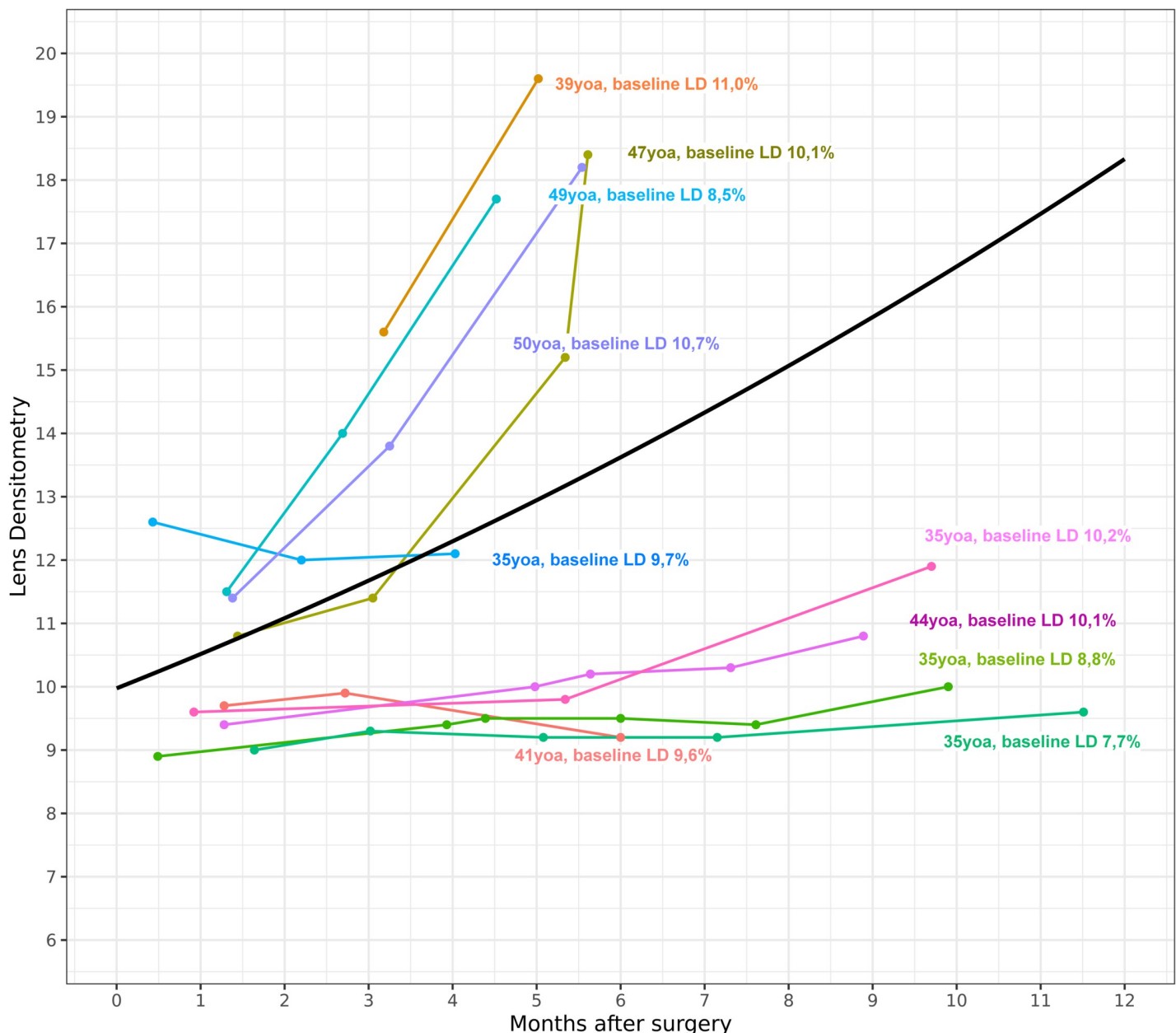

**Fig 3. Observed trajectories for patients under 51 years of age (yoa).** Changes in lens densitometry (%) after pars plana vitrectomy over time (months) for individual cases with younger age.

The number of cases in our study is still relatively small. Therefore, the decision for or against a combined phakovitrectomy cannot be completely based on the mathematical predictions.

In general, it is always the decision of the surgeon whether and when a cataract operation is performed. This decision is made on the basis of various clinical and epidemiological criteria. However, many of these criteria are subject to subjective perception. Additional objective parameters such as LD measurements might probably simplify and complement this decision in some cases. On the other hand, this mathematical model can be used to make predictions that also help the patient to understand why a combined phakovitrectomy would make sense or not.

## Conclusions

PPV will sooner or later lead to vision impairing cataract formation. Individual estimates of when and to what degree it will occur are difficult. Using objective parameters like LD values and patient's age allowed us to make precise predictions about the speed of lens opacification in silicone oil filled eyes after PPV for RRD. The accuracy of the model was especially obtained by giving baseline LD a higher weight in the equation than patient's age.

We developed a statistical model that can be accessed for free under the following link (https://statisticarium.com/apps/sample-apps/LensDensityOil/) to help vitreoretinal surgeons for consultation on the possibility to combine PPV with cataract surgery.

## Supporting information

**S1 File.**
(DOCX)

## Acknowledgments

Everyone who contributed to this work is listed as an author and meets the authorship criteria.

## Author Contributions

**Conceptualization:** Philipp Schindler, Vasyl Druchkiv, Christos Skevas.

**Data curation:** Philipp Schindler, Luca Mautone, Vasyl Druchkiv, Toam Katz.

**Formal analysis:** Vasyl Druchkiv.

**Investigation:** Philipp Schindler, Luca Mautone, Vasyl Druchkiv.

**Methodology:** Philipp Schindler, Vasyl Druchkiv.

**Project administration:** Christos Skevas.

**Software:** Vasyl Druchkiv.

**Supervision:** Martin Stephan Spitzer.

**Validation:** Philipp Schindler, Vasyl Druchkiv.

**Visualization:** Vasyl Druchkiv.

**Writing – original draft:** Philipp Schindler.

**Writing – review & editing:** Luca Mautone, Martin Stephan Spitzer, Christos Skevas.

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
