## [Decision Letter · Decision Letter 0]

19 Jan 2022

PONE-D-21-38031Predicting speed of progression of lens opacification after pars plana vitrectomy with silicone oilPLOS ONE

Dear Dr. Philipp Schindler,

Thank you for submitting your manuscript to PLOS ONE. After careful consideration, we feel that it has merit but does not fully meet PLOS ONE’s publication criteria as it currently stands. Therefore, we invite you to submit a revised version of the manuscript that addresses the points raised during the review process.

We look forward to receiving your revised manuscript.

Kind regards,

Xingjun Fan, PhD

Academic Editor

PLOS ONE

Journal Requirements:

Additional Editor Comments:

Both reviewers showed some favor of this study but at the same time raised significant concerns that require the authors to address accordingly. Notably, the sample size and exclusion criteria need to be carefully addressed.

Reviewers' comments:

Reviewer's Responses to Questions

**Comments to the Author**

1. Is the manuscript technically sound, and do the data support the conclusions?

Reviewer #1: Yes

Reviewer #2: Partly

2. Has the statistical analysis been performed appropriately and rigorously? 

Reviewer #1: Yes

Reviewer #2: Yes

3. Have the authors made all data underlying the findings in their manuscript fully available?

Reviewer #1: No

Reviewer #2: Yes

4. Is the manuscript presented in an intelligible fashion and written in standard English?

Reviewer #1: Yes

Reviewer #2: Yes

5. Review Comments to the Author

Reviewer #1: This is a good paper that adds clinical tools to the body of knowledge. Things that I noticed that are missing include how many different surgeons with listed experience level conducted these surgeries (you mention this to be an influential factor in the discussion) and intraoperative details (laser and amount used, time of surgery, fluid used, etc). A nice next step would be nice to apply this to other external data sets to see if the prediction model holds up.

Reviewer #2: The authors present a prospective study describing the predicting abilities of a regression model using Lens densitometry (LD) measurements of the PentacamHR® Nucleus Staging mode (PNS) and other parameters to estimate postoperative cataract formation after Pars Plana Vitrectomy (PPV) performed due to Rhegmatogenous Retinal Detachment (RRD).

The authors reported accurate predictions, with accuracy mainly attributed to the inclusion of LD baseline measurements, and to a lesser extent, the patient's age. The authors also provide a link in which doctors can use the suggested model for clinical application, which is appropriate and adds contribution. Data were properly made available.

The article is overall decently written. The statistical analysis performed was sound and sufficiently detailed.

However, unfortunately there are some flaws in this study that should to be addressed:

1) The authors have recently published a similar study, regarding C3F8 as opposed to silicone oil, which was indeed referenced in this study. It is hard to understand why the results of both studies were published separately. This causes a significant extent of this study to be replicated from the previous one. The authors should elaborate on this issue, as methods, results, and conclusions are also very similar.

2) The clinical correlation of the Lens densitometry (LD) is hardly addressed in this study. The decision to operate (if not performed simultaneously during PPV) is ideally made clinically, with respect to the patient's wishes. Do the LD measurements necessarily indicate the need for surgery, rather than the clinical presentation? The authors should elaborate on this premise. How this affects the surgeon's decision, to the reader, is left unclear.

3) The sample size may be too small to draw the aforementioned conclusions. Authors are advised to expand their sample size or address this limitation with more modest phrasing.

4) Can the authors provide information regarding the Variance inflation factors (VIFs) in the model?

5) It would be best to add a citation to the sentence in lines 257-259.

6) The authors should provide a reasonable explanation as to the major difference between the outcomes of different Lens densitometry (LD) baselines (10% vs 11%)? Could this be affected by other Confounding variables? Sample size?

7) The authors mention exclusion criteria by stating:

"Patients with prior cataract surgery on one eye, PPV, or any other intraocular procedure were excluded. Patients with history of ocular trauma, uveitis, topical or systemic corticosteroid therapy, and signs of visually impairing cataract at baseline evaluation were also excluded from the analysis. Additionally, known topical or systemic conditions (diabetes excluded) that could accelerate cataract formation and/or progression after PPV led to exclusion from the study. Patients with ischemic and/or proliferative retinopathy were also excluded" How many patients did this entail? Was this a fair amount that represents the cohort? What was the reasoning behind their exclusion criteria? The authors should also provide an explanation as to why Diabetes Mellitus was not excluded, as opposed to other conditions.

8) Do the authors have information on how long it took to perform the Lens densitometry (LD) measurements? Many readers might be interested in this, particularly those who have not clinically used the device.

9) Who performed the Lens densitometry (LD) measurements? Please provide details about the qualifications and background of this/these individual(s).

10) When giving the BCVA, could the authors also add the Snellen 20/X equivalents of the logMAR visual acuities listed?

11) The authors state: "PentacamHR® PNS module only depicts nuclear and no cortical or posterior subcapsular changes. PSC occurrence is described frequently under silicone oil and leads to considerable visual impairment quickly [36]. However, since PSC rarely appears isolated after silicone oil, but mostly in combination with NSC, we do not consider this weakness as a negative influence on the results of our study." Given the broad nature of readership of PLoS One, I think a supportive citation should be added.

12) The authors should add a proper "limitations" paragraph to their discussion section and address the various limitations. Are there no other significant limitations to the study besides the sole depiction of nuclear cataracts?

13) More clinical details on the patients would make this a stronger study, such as retinal and non-retinal co-morbidities. Given the small sample size, it is very important to understand the sample to know how applicable this to other patient populations. This should also be addressed in the limitations section.

14) Are there any data about the predictive value of other measures in the literature? It would be valuable to mention that in the discussion and provide relevant information, to get a sense of how well it does overall in comparison to other, perhaps more widely available, methods/instruments.

15) Due to the aforementioned, I think the conclusions drawn, especially in the way they were phrased – are too conclusive. It would best to use more cautious terminology.

6. PLOS authors have the option to publish the peer review history of their article (what does this mean?). If published, this will include your full peer review and any attached files.

Reviewer #1: No

Reviewer #2: No

---

## [Author Response · Author response to Decision Letter 0]

11 Mar 2022

Dear Reviewers,

First of all we want to thank you very much for your constructive criticism. We appreciate your comments and hope that we have answered your questions sufficiently clearly and satis-factorily.

Following you will find a list of your remarks and our corresponding answers.

Reviewer #1: 

Answer: Our plan is to spread the statistical model among other surgeons with access to Pen-tacamHR PNS measurements. When they plan to perform PPV without combined cataract surgery, we ask them to perform a PNS scan before surgery. Predictions by our model can be performed for any timepoint, for example one year after surgery. This can be made online on a freely accessible platform. After one year PNS scans should be repeated and compared to the calculated predic-tions.

Reviewer #2 remark 1): The authors have recently published a similar study, regarding C3F8 as op-posed to silicone oil, which was indeed referenced in this study. It is hard to understand why the results of both studies were published separately. This causes a significant extent of this study to be replicated from the previous one. The authors should elaborate on this issue, as methods, re-sults, and conclusions are also very similar.

Answer: Yes, statistical methods are the same. Dividing sample in training and test data sets -> mixed regression modelling with backward variable selection -> applying the resulted model on the test data set.

What essentially differs is the data set: In previous study the model is derived based on patients treated with gas and in the current study are patients treated with oil. 

Applying same methodology on different data may produce different result. As you can see the derived models are slightly different. In model for patients treated with C3F8-gas the polynomial term for month is retained. And in the model for silicone oil there is no polynomial for month but there is significant higher order interaction: Months X Age X Baseline_LD

Literature and clinical observations showed that silicone oil is thought to have different impact on lens opacification than C3F8 gas. It was our goal to get reliable data for patients treated with oil vs. patients treated with C3F8 gas. Developing one mathematical model from a dataset containing both groups, oil and gas, would impede the differentiation of their effect. The aim of both studies is to describe progression of lens opacification with objective parameter as precise as possible and from that data develop a statistical tool that adds clinical relevance for the understanding of cataract pro-gression and patient consultation for both types of endotamponade.

Reviewer #2 remark 2) and 3): 

Answer: Please see lines 340-352 in the revised manuscript with track changes

Reviewer #2 remark 4): Can the authors provide information regarding the Variance inflation fac-tors (VIFs) in the model?

Answer: It is not clear what you are implying. But, as expected, the VIFs for the parameters in the final model are very large. We should emphasize AS EXPECTED because we use interaction terms in our modelling. In models with interactions the large VIFs are not a problem and are expected. Fur-thermore the goal of the study is to make a prediction about densitometry trajectories and there-fore we are interested much more in the parameter estimates than in their standard errors. Recall that no matter how great the multicollinearity among a set of variables is, it in no way compromises the estimates associated with the other variables in the regression. Nevertheless we are willing to give you some idea about the correlation among the three main predictors in the model: age, month and baseline densitometry. If we estimate the model with only these free parameters without interaction among them we derive following Variance Inflation Factors: VIFmonth=1.005, VIFage=1.62 and VIFbaseline_density=1.61. These VIFs are fairly low to cause any 

concern.

Reviewer #2 remark 5): It would be best to add a citation to the sentence in lines 257-259.

Answer: We are not sure which sentences of the manuscript you are referring to, since line 257-259 already have several citations [21]–[24] and [19], [23], [25]–[28]. For a better aspect we added par-agraphs.

Reviewer #2 remark 6): The authors should provide a reasonable explanation as to the major dif-ference between the outcomes of different Lens densitometry (LD) baselines (10% vs 11%)? Could this be affected by other Confounding variables? Sample size?

Answer: Please see lines 298-314 in the revised manuscript with track changes. We rephrased this parts for a better clarification.

Reviewer #2: The authors mention exclusion criteria …How many patients did this entail? Was this a fair amount that represents the cohort? What was the reasoning behind their exclusion criteria? The authors should also provide an explanation as to why Diabetes Mellitus was not excluded, as opposed to other conditions.

Answer: In our clinic we approximately perform 1000 vitrectomies every year. The number of pa-tients presenting to our clinic with primary rhegmatogenous retinal detachment varies between 300-400 cases per year. These patients usually come as emergencies. Among those patients the majority is already pseudophakic. Remaining phakic cases (on both eyes) receive Pentacam PNS and IOL-Master measurements within our routine diagnostic procedure independent of whether PPV is combined with cataract surgery or not. This ensures that surgery will be able to start as soon as possible without any further delay due to missing diagnostic examinations. When the surgeon has made his or her decision which procedure will be performed, we screen those cases for inclu-sion. So, the clinical decision for or against phakocvitrectomy was made completely independent of this study. About 10 to 15% of the cases that would have met the criteria were lost because these cases arrived at the weekend and no PNS diagnostics were available. Included patients received verbal information about the upcoming measurements including Pentacam PNS during their rou-tine post-operative visits. An exact number of excluded patients cannot be given.

Diabetes mellitus (DM) was not excluded because of two reasons: First, DM is a frequently occur-ring condition in Europe and excluding those cases without any sings of diabetic retinopathy would lead to an even smaller cohort. Number two, diabetes is a known risk factor for cataract formation indeed, but if visually impairing cataract was present at the time of the first presentation of the patient, he or she would most likely have had combined phakovitrectomy. In addition, a recent meta-analysis showed that DM might not lead to nuclear sclerotic cataract (Li L, Wan XH, Zhao GH. Meta-analysis of the risk of cataract in type 2 diabetes. BMC Ophthalmol. 2014 Jul 24;14:94. doi: 10.1186/1471-2415-14-94. PMID: 25060855; PMCID: PMC4113025.) And some recent studies showed that DM could probably have some protective effect against cataract formation in vitrectomized eyes.

Reviewer #2 remark 8) and 9)

Answer: Please see lines 89-92 in the revised manuscript with track changes

Reviewer #2 remark 10): When giving the BCVA, could the authors also add the Snellen 20/X equivalents of the logMAR visual acuities listed?

Answer: units have been added in Table 1

Reviewer #2 remark 11) : Given the broad nature of readership of PLoS One, I think a supportive citation should be added.

Answer: We added a further citation for lines 340-343

Reviewer #2 remark 12) and 13):

Answer: Please see lines 344-346 and 350-359 in the revised manuscript with track changes

Reviewer #2 remark 14): Are there any data about the predictive value of other measures in the literature? It would be valuable to mention that in the discussion and provide relevant information, to get a sense of how well it does overall in comparison to other, perhaps more widely available, methods/instruments.

Answer: To our knowledge this and the predecessor study with C3F8-gas are the first studies that try to depict objective parameters for changes in lens status after PPV and, additionally, derive a mathematical model from that data to make future predictions for individual cases. Other models exist using artificial intelligence and deep learning to predict a risk for cataract from epidemiological data or to calculate the acuity of refractive power after cataract-surgery from preoperative measures.

Reviewer #2 remark 15) Due to the aforementioned, I think the conclusions drawn, especially in the way they were phrased – are too conclusive. It would best to use more cautious terminology.

Answer: We rephrased our conclusions according to the reviewer’s recommendations.

We want to thank you for your efforts.

Yours sincerely,

Dr. med. Phillip Schindler

Specialist for Ophthalmology

University Medical Center Hamburg-Eppendorf

---

## [Decision Letter · Decision Letter 1]

29 Apr 2022

Predicting speed of progression of lens opacification after pars plana vitrectomy with silicone oil

PONE-D-21-38031R1

Dear Dr. Philipp Schindler,

We’re pleased to inform you that your manuscript has been judged scientifically suitable for publication and will be formally accepted for publication once it meets all outstanding technical requirements.

Kind regards,

Xingjun Fan, PhD

Academic Editor

PLOS ONE

Additional Editor Comments (optional):

The authors are required to address the exclusion criteria.

Reviewers' comments:

Reviewer's Responses to Questions

**Comments to the Author**

1. If the authors have adequately addressed your comments raised in a previous round of review and you feel that this manuscript is now acceptable for publication, you may indicate that here to bypass the “Comments to the Author” section, enter your conflict of interest statement in the “Confidential to Editor” section, and submit your "Accept" recommendation.

Reviewer #1: All comments have been addressed

Reviewer #2: (No Response)

2. Is the manuscript technically sound, and do the data support the conclusions?

Reviewer #1: Yes

Reviewer #2: Partly

3. Has the statistical analysis been performed appropriately and rigorously? 

Reviewer #1: Yes

Reviewer #2: Yes

4. Have the authors made all data underlying the findings in their manuscript fully available?

Reviewer #1: Yes

Reviewer #2: Yes

5. Is the manuscript presented in an intelligible fashion and written in standard English?

Reviewer #1: Yes

Reviewer #2: Yes

6. Review Comments to the Author

Reviewer #1: (No Response)

Reviewer #2: A lot of the comments were addressed. Unfortunately, some remarks were brushed off, and major revisions were treated as minor ones.

1) The reply for the first inquiry was dismissive and non satisfactory. Of course, different data sets can produce different results using the same methodology, but this was not the issue raised.

2) Exclusion criteria was not propely addressed. In the case that no further details can be provided, this should be explained to the readers.

3) I believe that a bigger sample size and a broader inclusion criteria, along with co-morbidities and retinal co-morbidities (which are common with RD) would have made this a better study.

Other than the aforementioned, the authors have addressed the rest of the suggestions in a satisfactory fashion, and I believe their manuscript is better in its current form.

7. PLOS authors have the option to publish the peer review history of their article (what does this mean?). If published, this will include your full peer review and any attached files.

Reviewer #1: No

Reviewer #2: No

---

## [Editor Report · Acceptance letter]

12 May 2022

PONE-D-21-38031R1 

Predicting speed of progression of lens opacification after pars plana vitrectomy with silicone oil 

Dear Dr. Schindler:

I'm pleased to inform you that your manuscript has been deemed suitable for publication in PLOS ONE. Congratulations! Your manuscript is now with our production department. 

Kind regards, 

on behalf of

Dr. Xingjun Fan 

Academic Editor

PLOS ONE